# Development and Verification of a Deep Learning Algorithm to Evaluate Small-Bowel Preparation Quality

**DOI:** 10.3390/diagnostics11061127

**Published:** 2021-06-20

**Authors:** Ji Hyung Nam, Dong Jun Oh, Sumin Lee, Hyun Joo Song, Yun Jeong Lim

**Affiliations:** 1Division of Gastroenterology, Department of Internal Medicine, Dongguk University Ilsan Hospital, Dongguk University College of Medicine, Goyang 10326, Korea; drnamesl@gmail.com (J.H.N.); mileo31@naver.com (D.J.O.); lee.sumin3@gmail.com (S.L.); 2Division of Gastroenterology, Department of Internal Medicine, Jeju National University School of Medicine, Jeju 63241, Korea; hyun-ju2002@hanmail.net

**Keywords:** capsule endoscopy, quality of bowel preparation, deep learning algorithm, validation

## Abstract

Capsule endoscopy (CE) quality control requires an objective scoring system to evaluate the preparation of the small bowel (SB). We propose a deep learning algorithm to calculate SB cleansing scores and verify the algorithm’s performance. A 5-point scoring system based on clarity of mucosal visualization was used to develop the deep learning algorithm (400,000 frames; 280,000 for training and 120,000 for testing). External validation was performed using additional CE cases (*n* = 50), and average cleansing scores (1.0 to 5.0) calculated using the algorithm were compared to clinical grades (A to C) assigned by clinicians. Test results obtained using 120,000 frames exhibited 93% accuracy. The separate CE case exhibited substantial agreement between the deep learning algorithm scores and clinicians’ assessments (Cohen’s kappa: 0.672). In the external validation, the cleansing score decreased with worsening clinical grade (scores of 3.9, 3.2, and 2.5 for grades A, B, and C, respectively, *p* < 0.001). Receiver operating characteristic curve analysis revealed that a cleansing score cut-off of 2.95 indicated clinically adequate preparation. This algorithm provides an objective and automated cleansing score for evaluating SB preparation for CE. The results of this study will serve as clinical evidence supporting the practical use of deep learning algorithms for evaluating SB preparation quality.

## 1. Introduction

Capsule endoscopy (CE) is an imaging tool that allows direct observation of the entire small bowel (SB) [1]. Since a 2007 statement by American Gastroenterology Association proposed that CE should be the third test following standard endoscopy (esophagogastroduodenoscopy and colonoscopy) in patients with obscure gastrointestinal (GI) bleeding [2], guidelines recommend it as the initial diagnostic modality when obscure GI bleeding, Crohn’s disease, SB tumors, or polyposis syndrome are suspected [3]. CE also carries a low risk of sedation-related complications and is minimally invasive [4]. In addition, its coverage is expanding to other GI areas, such as the colon and stomach [5,6], as an alternative to conventional endoscopy and for screening purposes. Endoscopic examination of the lower GI tract, including the SB, requires fasting and proper bowel preparation using purgative agents. As such, sufficient bowel cleansing is one of the most important factors affecting the interpretation and reliability of endoscopic findings. Because CE involves retrospective interpretation of previously obtained and transmitted video images, the modality depends more on the quality of bowel preparation than conventional endoscopy which allows active cleansing during examination. In clinical practice, the SB mucosa is often found to be obscured by air bubbles or residual material in the intestine during CE reading. The more the SB mucosa is obscured, the more likely it is that lesions will be missed. Therefore, suboptimal bowel preparation reduces the reliability of CE results, resulting in repeat examinations and additional costs [7].

Current guidelines recommend that patients ingest 2 L of polyethylene glycol (PEG) for cleansing prior to SB CE to facilitate mucosal visualization [8]. However, even after administration of appropriate purgative agents, the degree of mucosal visualization resulting in the actual reading images can differ depending on each patient’s clinical characteristics, such as age, physical activity, underlying diseases, and bowel movement. Therefore, it is recommended that the adequacy of bowel preparation be described in the CE report [3]. However, no standardized scales are available for evaluating the quality of SB cleansing. It is virtually impossible for a CE reader to objectively determine the quality of bowel cleansing for tens of thousands of images taken over an average of 6 h. Several grading scales have been reported in the literature and used clinically [7,9,10], but these scales are not objective because the evaluation depends on visual assessments made by individual CE readers. Several automated calculation software programs employing color intensity that can be integrated into CE reading programs have been recently introduced [11,12]. However, these programs are not suitable for clinical use because the condensed color bands are difficult to represent across entire SB images, and it is difficult to distinguish color changes due to obscuration and lesion-associated color changes, such as bleeding. Therefore, it is necessary to develop an algorithm that can automatically evaluate the quality of bowel preparation. A potentially worthwhile approach is to utilize deep learning, which has recently been attempted in several endoscopic applications. The objective of this study was to develop a deep learning-based automatic calculation algorithm for evaluating SB preparation quality and verify its clinical applicability.

## 2. Materials and Methods

### 2.1. Study Design

The study included SB CEs (MiroCam MC-1000, MC-1200, MC-4000, Intromedic Ltd., Seoul, Korea) performed consecutively at two university hospitals (Dongguk University Ilsan Hospital and Jeju National University Hospital) between March 2012 and September 2020. Reasons for CE examinations included obscure GI bleeding, suspected or established Crohn’s disease, and suspected SB tumor or polyposis. All patients were instructed to fast overnight and ingest 2 L of PEG plus ascorbic acid (Coolprep, Taejoon Pharm. Co., Seoul, Korea) prior to CE. Incomplete CEs, in which the cecum was not reached due to capsule retention or power limitation, and inaccessibility of videos due to mechanical errors were excluded from the study. Among eligible cases, 100 CEs performed at Dongguk University between March 2012 and June 2020 were selected for the development of the deep-learning database (training set) (Figure 1). Separate validation was performed using a total of 50 different CEs from the two hospitals (Jeju, *n* = 33; Dongguk, *n* = 17) (validation set). The study was conducted in accordance with the guidelines of the Declaration of Helsinki and approved by the Institutional Review Board of Dongguk University Ilsan Hospital (IRB no. DUIH 2020-08-018).

### 2.2. Materials

Significant abnormalities, such as extensive bleeding and large ulcers, could confuse training regarding the status of bowel cleansing. Thus, after excluding video segments exhibiting such abnormalities, 400,000 still-cut frames were extracted from the training set using an optical character recognition (OCR) program. The degree of bowel cleansing was scored from 1 to 5, according to the proportion of the mucosa that was not obscured by air bubbles, bile, or debris in each frame (Figure 2) [13], with a higher score indicating a greater degree of cleansing: Score of 5 (>90% of mucosa visible), score of 1 (<25% of mucosa visible). Two experienced CE readers (J.H.N. and D.J.O.) determined the cleansing score for the 400,000 frames according to the 5-point scoring scale. If there were any discrepancies between the readers’ scores, the final score was determined after re-evaluation and discussion with a senior reader (Y.J.L.). The 400,000 frames assigned to one of the five scoring categories were randomly divided into groups of 280,000 and 120,000 frames, respectively, and used for training and testing.

### 2.3. Training and Testing

The deep learning algorithm development process consisted of pre-processing of input data and repetition of training and testing, as follows.

#### 2.3.1. Pre-Processing of Input Data

Training was conducted using the characteristics of texture, color, and shape. Because training results can be affected by color, processing of images (e.g., manipulation of white balance) was required to minimize color changes. In the RGB, HSV, and Lab color models, the Gray, HSV-S, and Lab-b components, respectively, were used as input data for training: Type 1 (RGB-Gray), type 2 (HSV-S), and input type 3 (Lab-b) (Figure 3a).

#### 2.3.2. Repetition of Training and Testing

The deep learning algorithm was constructed based on the CNN (convolutional neural network) model, the most widely used model for image recognition (Figure 3b). A combination of values between 0 and 1 for input types 1, 2, and 3 was used to determine the cleansing score for the frames. Training was started with 0.001 for learning rate, and full layers were then trained with 0.00001 for learning rate. Because the dataset was also inevitably classified based on the clinician’s subjective evaluation, uncertainty between two adjacent scores was allowed. The optimizer used the RMSProp technique. Prior to hard determination of the score, the probability for each score was predicted by applying the softmax function and loss function of categorical cross-entropy. The score with the highest probability is assigned as the cleansing score of the image as:$$S≜\underset{p(Gi)}{\max}Gi$$
where S is score of the image, max is maximum, *Gi* is *i*-grade, 1 ≤ *i* ≤ 5, and *P*(*Gi*) represents probability of *i*-grade for the image. The degree of agreement between the score with the highest probability in the algorithm and the score classified by CE readers was then evaluated (Top-1 accuracy). Agreement between the score calculated using the algorithm and the score determined by the CE reader was confirmed for all SB frames in a separate CE case. The average score of all images was calculated as the final cleansing score for each CE as:(1)$$Final\;score=\frac{1}{N}\overset{N}{\underset{n=0}{\sum }}Sn$$
where N indicates number of total images, and S is score of the image.

### 2.4. Validation Set

The algorithm was validated using 50 CE cases that were separate from the training set. All video segments corresponding to SB sections in the validation set were extracted into frames using the OCR program. Extracted frames were divided into three groups with an equal number of segments according to the video time sequence: segment 1 (seg1), proximal third; segment 2 (seg2), middle third; and segment 3 (seg3), distal third. Average cleansing scores per validation case were then calculated using the developed algorithm. Two CE readers who were blinded to the cleansing scores calculated using the algorithm determined clinical grading (overall grading A to C) using a previously validated bowel preparation scale [14]. The scale employed a quantitative parameter based on the proportion of non-prepped images in which air bubbles, bile, and debris resulted in >50% disruption of visualization (Table 1). An overall grading of A or B was classified as clinically adequate bowel preparation, whereas an overall grading of C was considered inadequate bowel preparation. Disagreement between the two readers was resolved through discussion with the senior reader. Finally, the average cleansing score calculated using the algorithm and clinical grading determined by the CE reader were compared.

### 2.5. Methodology

The primary study outcome was the performance of the algorithm for assessing the quality of SB preparation. Top-1 accuracy was determined by dividing the number of concordant pairs by 120,000. In addition, for a separate case, agreement between the score was calculated using the algorithm and the score was determined by the CE reader was evaluated using Cohen’s kappa value:K = *P_o_* − *P_e_*/1 − *P_e_*(2)
where *P_o_* indicated the number in agreement divided by total pairs, and *P_e_* indicated the sum of the percentage of agreement due to chance per each score. Next, the average cleansing score (from 1.0 to 5.0) per validation case was calculated as the sum of cleansing scores divided by the number of frames. Analysis of variance was used to compare average cleansing scores among the different overall grading (A to C) groups. A post hoc analysis was performed using Dunnett’s test. Differences in average cleansing scores for the clinically adequate and inadequate preparation groups were compared using independent sample *t*-tests. The sensitivity and specificity for determining clinically adequate preparation were calculated for each average cleansing score (1.0 to 5.0). A receiver operating characteristic (ROC) curve was generated to determine the cleansing score cut-off value for clinically adequate preparation. Two-sided *p* values of <0.05 were considered statistically significant. All statistical analyses were conducted using SPSS Statistics, version 19.0 (IBM, Armonk, NY, USA).

## 3. Results

### 3.1. Performance Evaluation

Testing using 120 frames within the training set resulted in a Top-1 accuracy of 93%. For analysis of 51,380 total frames from a separate CE case from Dongguk University, substantial agreement was observed between the cleansing scores determined using the deep learning algorithm and the clinician’s assessment (misclassification rate, 24.7%; Cohen’s kappa value, 0.672). The underestimation rate was 19.2%, which was more common than the overestimation rate of 5.5%. The mean cleansing scores determined using the algorithm and clinician’s assessment were 2.9 and 3.1, respectively (Table 2).

### 3.2. External Validation and Cut-Off Value

The mean patient age among the 50 validation cases was 45.0 ± 23.3 years (range, 10–84 years). The validation group included 34 (68.0%) males. Mean SB transit time was 5.6 ± 2.3 h (range, 1.7–11.0 h). A total of 11 (22.0%), 20 (40.0%), and 19 (38.0%) cases were classified as overall image quality grades A, B, and C, respectively. Average cleansing scores decreased with worsening overall quality grades from A to C, with scores of 3.9 ± 0.3, 3.2 ± 0.5, and 2.5 ± 0.5 for grades A, B, and C, respectively (Table 3, Figure 4a). Grades A and B exhibited significantly higher average cleansing scores than grade C (both *p* < 0.001).

Adequate preparation was achieved in 62% (31/50) of cases in the validation set. The average cleansing score for the adequate preparation group was significantly higher than that for the inadequate preparation group (3.4 vs. 2.5, *p* < 0.001). ROC curve analysis indicated a cleansing score cut-off value of 2.95 for clinically adequate preparation, with a sensitivity of 81%, specificity of 84%, and area under the curve of 0.913 (95% confidence interval, 0.835–0.990, *p* < 0.001) (Figure 4b).

## 4. Discussion

The development of this new algorithm for automated calculation of SB cleansing score was based on an objective classification system for mucosal visibility using a simple deep learning method and a total of 400,000 CE images. Testing of the algorithm revealed a high Top-1 accuracy of 93%. We also confirmed substantial agreement between cleansing scores determined using the developed algorithm and those determined by CE readers in an analysis of a separate case. In addition, the algorithm exhibited good performance in determining SB preparation quality in validation testing. As there is no standard deep learning approach for classification of SB cleansing quality, it can be difficult to judge the importance of the developed algorithm. Therefore, it is critical to compare the algorithm with the clinical grades that are already in use by clinicians. We focused on whether the average cleansing score calculated by the algorithm could simply represent the adequacy of SB cleansing quality.

Recently, computer-aided assessments have been introduced for evaluating colon cleansing, providing objective calculation of bowel preparation for colonoscopy [15,16]. Even though the clinical needs are greater for SB than for colon, the previously proposed CNN models provided a basis for the development of SB algorithm in the current study. They applied a widely used 4-level Boston preparation cleansing scale based on the amount of fecal residue. Meanwhile, score 5 (more than 90% mucosa are visible) was separately classified in our study because we needed to train completely cleaned mucosa. We previously conducted a preliminary study using PillCam CE (PillCam SB3, GIVEN Imaging Ltd., Yokneam Illit, Israel) [13], which included only 3500 frames for training set. In detail, 700 frames per each cleansing score (1 to 5) were selected in consideration of the diversity of images with regard to factors such as bile, air bubbles, debris, and the diversity between patient cases and used for deep learning. As a result, it exhibited a Top-1 accuracy of 69.4%. In the current study using MiroCam CE, an enormous image of 280,000 frames was used for training to increase the accuracy of the deep learning model. Subsequently, not only were more diverse images trained, but similar images of the same cases were also repeatedly trained, resulting in improved accuracy of the training set. Meanwhile, the Top-2 accuracy (including the first and second highest probabilities) was as high as 90% in the previous PillCam study as well. In addition, a separate case that did not overlap with the training set exhibited a 24.7% misclassification rate in the current study, lower than the test result for the training set. Although the accuracy varied slightly depending on the study design and testing like this, external validation testing revealed that automatically calculated cleansing scores can be used to clinically determine bowel preparation quality. The adequacy of bowel preparation is dependent upon the overall quality of each frame, but accurately classifying the score of every frame is not necessary in clinical practice. Instead, the calculated cleansing score should be suitable for determining whether bowel preparation was clinically adequate. Both studies are of clinical significance in that a cleansing score cut-off value for adequate bowel preparation was suggested. The cut-off value exhibited lower sensitivity and specificity in the present study than in the preliminary study. This discrepancy may be related to the difference in the number of cases used for validation between the studies. To obtain a useful algorithm, it is also considered important to train using an adequate number and variety of images from multiple patients, as might be observed in clinical practice.

With the expansion of CE indications and increasing demand, various technical improvements have been made, such as 3D reconstruction of SB lesions, super-resolution, and active control of the capsule via magnetic assistance [3,17,18,19]. In addition, research into artificial intelligence (AI) models that enable effective detection of SB lesions is increasing [20,21,22,23]. The authors recently developed an AI-assisted reading model that determines the clinical significance of CE images [24]. The AI model significantly reduced the reading time and improved the efficiency of CE image reading, especially for trainees. Another interesting approach to reduce CE reading time was recently introduced. They proposed a CE frame reduction system based on a color structural similarity using color models for color texture and structural information, achieving a reduction ratio of 93.8% [25]. Although technical advances such as AI are expected to enhance the diagnostic yield of CE, there are some challenges relating to the SB preparation quality. Diagnostic yield is greatly affected by the quality of bowel preparation. Suboptimal bowel cleansing not only decreases diagnostic yield by impeding mucosal visualization, it also leads to incomplete examinations due to capsule retention and increased SB transit time [26,27]. Current guidelines specify bowel preparation quality as an important performance measure for SB CE. However, no single standardized bowel preparation scale is available [3]. In order to overcome this limit, we developed an automated device that can calculate SB cleanliness. The use of the AI model for automated calculation of the SB cleansing score can provide an objective standardized scale for evaluating bowel preparation quality, thereby also enabling evaluation of whether the CE examination was appropriate and its results reliable. Accordingly, the algorithm described here could help in determining the need for repeat examination or further treatment and could also serve as an indicator of CE quality control. It would be interesting to evaluate the diagnostic yield in examinations determined as insufficient preparation based on the algorithm. Moreover, additional advances in the algorithm are expected as more CE case experiences are integrated in the future. The developed algorithm can be loaded as a user interface in the review mode of the CE reading system in clinical practice, which confirms the clinical accessibility of the algorithm (Appendix A).

## 5. Conclusions

The exponential advancement of the computational capacity, with a greater understanding of deep learning methods, has greatly improved the ability to predict and improve SB imaging, and the need for scoring to improve bowel preparation has emerged. We developed a deep learning algorithm that automatically calculates the SB cleansing score, thus providing an objective scale to evaluate the quality of bowel preparation. External validation demonstrated good performance of the developed algorithm using a previously validated preparation scale that is currently used clinically. The proposed cleansing score cut-off value may serve as a standard quality indicator for CE. The results of this study provide clinical evidence supporting the practical use of deep learning algorithms for evaluating SB preparation quality.

## Figures and Tables

**Figure 1 diagnostics-11-01127-f001:**
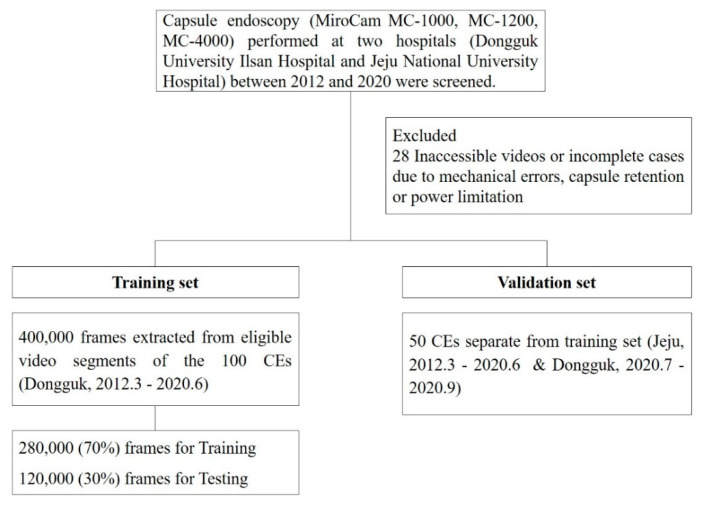
Data flow. CE, capsule endoscopy.

**Figure 2 diagnostics-11-01127-f002:**
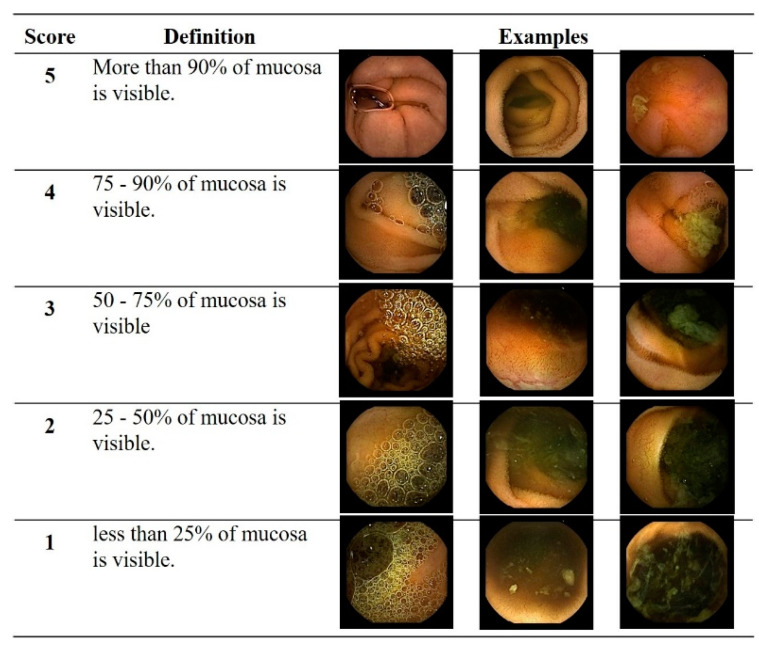
Cleansing score used for deep learning: A 5-point scoring method based on the proportion of mucosa visible.

**Figure 3 diagnostics-11-01127-f003:**
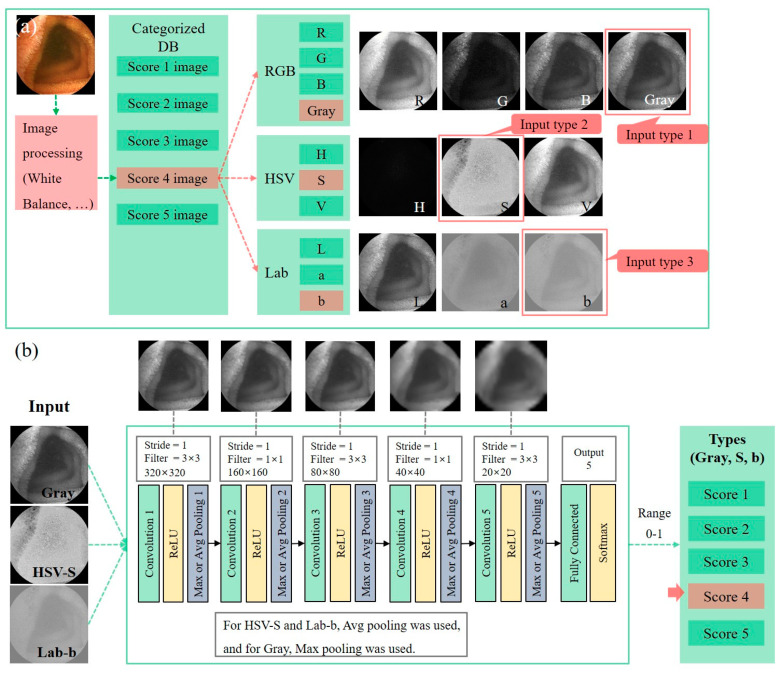
Deep learning process for algorithm development. (**a**) Pre-processing of input data, (**b**) repeat of training and testing. DB, database; HSV, hue saturation value; ReLU, rectified linear unit; RGB, red green blue.

**Figure 4 diagnostics-11-01127-f004:**
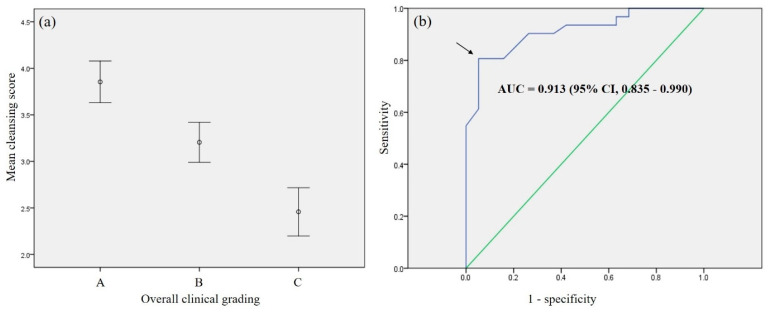
Outcomes of external validation. (**a**) Distribution of clinical grading and average cleansing scores of the validation cases; (**b**) Receiver operating characteristic curve of average cleansing score for clinically adequate preparation, indicating an estimated cut-off value of 2.95 (arrow). AUC, area under the curve.

**Table 1 diagnostics-11-01127-t001:** Validated small bowel preparation scale using a quantitative parameter.

Segmental Grading (Mucosal Invisibility of Each Segment)	
Grade 1	<5% of video images exhibiting >50% invisible mucosa due to air bubbles, bile, or debris
Grade 2	5–15%
Grade 3	15–25%
Grade 4	>25%
**Overall grading (overall cleansing quality)**	
Grade A	Total grades 3–5
Grade B	6–8
Grade C	9–12
**Clinically adequate preparation**	
Adequate	Grade A or B
Inadequate	Grade C

**Table 2 diagnostics-11-01127-t002:** The performance of deep learning algorithm calculating small bowel cleansing score, agreement with clinician’s assessment.

	51,380 Frames, *n* (%)					Mean Score
	S1	S2	S3	S4	S5	
Deep learning	3227 (6.3)	17,876 (34.8)	16,264 (31.7)	9354 (18.2)	4659 (9.1)	2.9
Clinicians	3761 (7.3)	16,696 (32.5)	14,723 (28.7)	4603 (9.0)	11,597 (22.6)	3.1

S1, <25% of mucosa visible; S2, 25–50%; S3, 50–75%; S4, 75–90%; S5, >90%.

**Table 3 diagnostics-11-01127-t003:** Average cleansing scores according to overall clinical grading of validation cases.

Overall Grading	*n* (%)	Score, Mean ± SD	95% CI	*p* Value *
A	11 (22.0)	2.9 ± 0.3	3.6–4.1	<0.001
B	20 (40.0)	3.2 ± 0.5	3.0–3.4	<0.001
C	19 (38.0)	2.5 ± 0.5	2.2–2.7	Ref.

CI, confidence interval; SD, standard deviation. * *p* values for post hoc analysis.

## Data Availability

Not available.

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
