# Peer review of "Development and Verification of a Deep Learning Algorithm to Evaluate Small-Bowel Preparation Quality"

_diagnostics, 2021, doi:10.3390/diagnostics11061127_

Round 1

Reviewer 1 Report

This paper presents a deep learning (DL) approach to evaluate bowel preparation quality. Although the paper is well written, there are important concerns, especially from the point of view of the technical contributions, as follows:

- There is no review of other DL methods previously proposed for similar tasks. It is therefore hard to judge if the proposed DL approach is novel or has any important technical contributions. A simple search for related works reveals a few previous published papers that are relevant, for example:

(a) Pogorelov, Konstantin, et al. "Nerthus: A bowel preparation quality video dataset." Proceedings of the 8th ACM on Multimedia Systems Conference. 2017.

(b) Zhou, Jie, et al. "A novel artificial intelligence system for the assessment of bowel preparation (with video)." Gastrointestinal endoscopy 91.2 (2020): 428-435.

- The proposed DL method is pretty simplistic. It uses well-known CNNs. There is not much novelty in this approach.

- It is not clear how the test and validation sets are used. Fig. 1 shows that the testing set is part of what is called the 'training set' Hence, this raises some concerns as to whether the test set was used at all during training. If so, then the model is not trained properly. Fig. 1 also shows a 'Validation set'. How is this used? Validation sets are usually a subset of the training set and are used to fine-tune hyperparameters of the DL model during training before testing it with the testing set. The way the datasets were used is very confusing. It is not clear if they were used as expected in any machine\deep learning model.

- In Fig. 3 (b), it is not clear why the output layer has 10 neurons if only 5 classes are expected (one per score).

- No details of how the network was optimized (stochastic gradient descent?) or the type of loss function used are given.

- Since the method is not compared against any other DL approach or standard machine learning approach for classification, it is hard to judge if the result obtained by the model are indeed important.

Author Response

This paper presents a deep learning (DL) approach to evaluate bowel preparation quality. Although the paper is well written, there are important concerns, especially from the point of view of the technical contributions, as follows:

  1. There is no review of other DL methods previously proposed for similar tasks. It is therefore hard to judge if the proposed DL approach is novel or has any important technical contributions. A simple search for related works reveals a few previous published papers that are relevant, for example:

(a) Pogorelov, Konstantin, et al. "Nerthus: A bowel preparation quality video dataset." Proceedings of the 8th ACM on Multimedia Systems Conference. 2017.

(b) Zhou, Jie, et al. "A novel artificial intelligence system for the assessment of bowel preparation (with video)." Gastrointestinal endoscopy 91.2 (2020): 428-435.

Response to comment) Thanks for the critical and thoughtful comments. The suggested papers are related with bowel preparation for colonoscopy. We added the references in the discussion section as follows,

[2nd paragraph of Discussion]

Recently, computer-aided assessments have been introduced for evaluating colon cleansing, providing objective calculation of bowel preparation for colonoscopy [15,16]. Even though the clinical needs are greater for SB than for colon, the previously proposed CNN models provided a basis for the development of SB algorithm in the current study. They applied widely used 4-level Boston preparation cleansing scale based on the amount of fecal residue. Meanwhile, score 5 (more than 90% mucosa are visible) was separately classified in our study because we needed to train completely cleaned mucosa. We previously conducted a preliminary study using PillCam CE (PillCam SB3, GIVEN Imaging Ltd., Yokneam Illit, Israel) [13], which included only 3,500 frames for training set.

  1. Pogorelov, K.; Randel, K.R.; de Lange, T.; Eskeland, S.L.; Griwodz, C.; Johansen, D.; Spampinato, C.; Taschwer, M.; Lux, M.; Schmidt, P.T.; Riegler, M.; Halvorsen, P. Nerthus: A bowel preparation quality video dataset, Proceedings of the 8th ACM on Multimedia Systems Conference; ACM: Taipei, 2017; pp. 170-174.

16. Zhou, J.; Wu, L.; Wan, X.; Shen, L.; Liu, J.; Zhang, J.; Jiang, X.; Wang, Z.; Yu, S.; Kang, J.; Li, M.; Hu, S.; Hu, X.; Gong, D.; Chen, D.; Yao, L.; Zhu, Y.; Yu, H. A novel artificial intelligence system for the assessment of bowel preparation (with video). Gastrointest. Endosc. 2020, 91, 428-435.

  1. The proposed DL method is pretty simplistic. It uses well-known CNNs. There is not much novelty in this approach.

Response to comment) We agree with the reviewer’s comment. As mentioned in the methods section, we used a popular CNN model for image recognition. It is significant that we have developed for the first time a clinically useful algorithm to evaluate small-bowel preparation quality in MiroCam CE using a simple deep learning method. We revised sentences in the discussion section as follows,

[1st sentence of Discussion]

Development of this new algorithm for automated calculation of SB cleansing score was based on an objective classification system for mucosal visibility using a simple deep learning method and a total of 400,000 CE images.

  1. It is not clear how the test and validation sets are used. Fig. 1 shows that the testing set is part of what is called the 'training set' Hence, this raises some concerns as to whether the test set was used at all during training. If so, then the model is not trained properly. Fig. 1 also shows a 'Validation set'. How is this used? Validation sets are usually a subset of the training set and are used to fine-tune hyperparameters of the DL model during training before testing it with the testing set. The way the datasets were used is very confusing. It is not clear if they were used as expected in any machine\deep learning model.

Response to comment) Thanks for the question. We understood the reviewer’s concerns. However, as shown in Figure 1, 70% (280,000 images) of training set was used for training, and remaining 120,000 images were used for testing. Thus, they are not overlapped. In addition, ‘Validation set’ is not a subset of the training set in our study. It consisted of independent CE cases and was not rated by clinicians on a scale of S1~S5. We just compared the average of the algorithm’s score according to clinical grading scale in “Validation set”. The clinical grading scale is classified as A, B, or C, which is a currently used cleansing scale in clinical practice.

  1. In Fig. 3 (b), it is not clear why the output layer has 10 neurons if only 5 classes are expected (one per score).

Response to comment) We appreciate the reviewer’s comment. It is a typo in the process of changing the deep learning model. In Fig.3 (b), output 10 was corrected to 5.

  1. No details of how the network was optimized (stochastic gradient descent?) or the type of loss function used are given.

Response to comment) According to the reviewer’s comments, we revised the sentences in Methods section as follows,

[4th paragraph of Methods]

2.3.2. Repetition of training and testing

The deep learning algorithm was constructed based on the CNN (convolutional neural network) model, the most widely used model for image recognition (Figure 3b). A combination of values between 0 and 1 for input types 1, 2, and 3 was used to determine the cleansing score for the frames. Training was started with 0.001 for learning rate, and full layers were then trained with 0.00001 for learning rate. Because the dataset was also inevitably classified based on the clinician’s subjective evaluation, uncertainty between two adjacent scores was allowed. The optimizer used the RMSProp technique. Prior to hard determination of the score, the probability for each score was predicted by applying the softmax function and loss function of categorical cross entropy.

  1. Since the method is not compared against any other DL approach or standard machine learning approach for classification, it is hard to judge if the result obtained by the model are indeed important.

Response to comment) We agree with the reviewer’s comment. There is no standard deep learning approach for classification of small-bowel cleansing. We added related sentences in the discussion section as follows,

[1st paragraph of Discussion]

As there is no standard deep learning approach for classification of SB cleansing quality, it can be difficult to judge the importance of the developed algorithm. Therefore, it is critical to compare the algorithm with the clinical grades that are already in use by clinicians. We focused on whether the average cleansing score calculated by the algorithm could simply represent the adequacy of SB cleansing quality.

Reviewer 2 Report

In this article, the authors report a deep learning algorithm to evaluate SB preparation quality. 

  1. The overall quality of the paper is well written, precise, crisp and to the point. Enjoyed reading it
  2. Abstract: Written well
  3. Images: Figure 2→ The figure provided has been reported in another journal by Nam et al. in scientific reports. Refer to Nam, J.H., Hwang, Y., Oh, D.J. et al. Development of a deep learning-based software for calculating cleansing score in small bowel capsule endoscopy. Sci Rep 11, 4417 (2021). Please make sure there are no copyright issues related to this. 
  4. Methods, the discussion is written well. The exponential development of the computational capacity of computers, with a greater understanding of deep learning methods, has greatly improved our ability to predict, improve the SB imaging, and need for scoring to improve bowel prep. This study certainly an effort towards it.

Reviewer 3 Report

In the present original article Nam et al elaborated a system, based on deep learning, to evaluate the quality of small bowel preparation for video capsule endoscopy (VCE). Compared to clinical judgment of a physician with experience in VCE, this algorithm had 93% accuracy and an agreement score of 0.672. Main comments:

1) In table 2, please explain the meaning of numbers under the columns of S1 - - > S5.

2) In how many cases did the algorithm overestimate or underestimate the quality of bowel preparation compared to human readers?

3) Despite this was not within the aims of the research, it is fundamental to underline that the quality of preparation correlates with VCE diagnostic yield. Therefore it would have been interesting to compare the diagnostic yield in examinations with insufficient preparation in algorithm versus human readers.
